# Safety Monitoring of mRNA COVID-19 Vaccines in Children Aged 5 to 11 Years by Using EudraVigilance Pharmacovigilance Database: The CoVaxChild Study

**DOI:** 10.3390/vaccines11020401

**Published:** 2023-02-09

**Authors:** Alessia Zinzi, Mario Gaio, Valerio Liguori, Rosanna Ruggiero, Marina Tesorone, Francesco Rossi, Concetta Rafaniello, Annalisa Capuano

**Affiliations:** 1Campania Regional Centre for Pharmacovigilance and Pharmacoepidemiology, 80138 Naples, Italy; 2Section of Pharmacology “L. Donatelli”, Department of Experimental Medicine, University of Campania “Luigi Vanvitelli”, 80138 Naples, Italy; 3Local Health Unit, ASL Napoli 1 Centro, 80145 Naples, Italy

**Keywords:** AEFI, COVID-19 vaccines, EudraVigilance, pharmacovigilance, safety, paediatric population, multisystem inflammatory syndrome, SARS-CoV-2, children

## Abstract

Although the safety profiles of mRNA COVID-19 vaccines (mRNA-1273 and BNT162b2) were evaluated in pre-authorization clinical trials, real-world data allow us to better define their benefit/risk ratio in the paediatric population. The current study aimed to evaluate the safety profiles of mRNA COVID-19 vaccines in children by analysing the pharmacovigilance data of the European spontaneous reporting system database EudraVigilance (EV) in the period from 1 January 2021, to 1 October 2022. During our study period, overall 4838 ICSRs related to mRNA COVID-19 vaccines referring to 5–11-year-old subjects were retrieved from EV, of which 96.9% were related to BNT162b2 and 49.3% were related to males. A total of 12,751 Adverse Events Following Immunization (AEFIs) were identified, of which 38.7% were serious. The most frequently reported AEFIs were pyrexia, headache, and vomiting. Only 20 Individual Case Safety Reports (ICSRs) reported Multisystem Inflammatory Syndrome (MIS) as an AEFI, all related to BNT162b2. The majority of MIS cases were females, and six cases were completely resolved at the time of reporting. Our results show a favourable risk–benefit profile for all mRNA COVID-19 vaccines in this paediatric sub-population, supporting their use in children. Considering the peculiarity and fragility of children, continuous safety monitoring of COVID-19 vaccines is required.

## 1. Introduction

Vaccines are fundamental tools to contrast preventable infectious diseases, including COronaVIrus Disease (COVID-19) due to the new COronaVIrus SARS-CoV-2 (Severe Acute Respiratory Syndrome COronaVIrus 2). Anti-COVID-19 vaccines have been fundamental weapons in contrasting the pandemic. Mass vaccination programs allowed to reduce the total number of COVID-19-related severe cases, hospitalizations, and deaths. In Europe, the authorized vaccines against COVID-19, until September 2022, included the following: two mRNA-platform-based vaccines, BNT162b2 (Pfizer-BioNTech) and mRNA-1273 (Moderna), consisting of lipid particles with the genetic information necessary for the synthesis of spike protein; two viral vector vaccines, ChAdOx1-S and Ad26.COV2-S, which use an adenovirus (unable to perform replication) to bring within the cell the sequence of the genetic code encoding for the spike protein; and a protein vaccine Nuvaxovid, containing a recombinant spike of the original SARS-CoV-2 virus produced in the laboratory. Recently, in June 2022, the inactivated, adjuvanted vaccine Valneva has been authorized by the European Medicine Agency (EMA) for protecting people aged between 18 and 50 years against COVID-19. During the pandemic, the use of some of the above-mentioned vaccines was extended to the paediatric population, with specific formulations (Figure 1). Specifically, BNT162b2, mRNA-1273, and Nuvaxovid have been authorized by the EMA for vaccination in the paediatric population in Europe. BNT162b2 was initially extended by the EMA in paediatric groups 12–15 (in May 2021) and then for 5–11 years old (in November 2021) [1]. Similarly, in July and November 2021, EMA authorized the mRNA-1273 use for vaccination of subjects aged 12–17 and >5 years old, respectively [2]. In June 2022, Nuvaxovid authorization was also extended to include individuals ≥12 years of age. Recently, paediatric formulations of BNT162b2 and mRNA-1273 have also been authorized for children ages 6 months.

Regulatory agencies and main international and national paediatric scientific societies promote the COVID-19 vaccination in children over 5 years to prevent severe disease and avoid long-term consequences of virus infection among the youngest [3]. However, paediatric COVID-19 vaccination is still debated, and it is associated with an important parental hesitancy [4]. This is mainly due to the higher rate of asymptomatic or paucisymptomatic cases among paediatric groups compared with the adult population [5]. Indeed, paediatric patients affected by SARS-CoV-2 infection show generally mild signs and symptoms [6,7]. Although more rarely, severe COVID-19 cases can also occur in the paediatric population [8]. In some cases, paediatric COVID-19 can be associated with rare but serious consequences, e.g., the risk of developing Multisystem Inflammatory Syndrome in children (MIS-c), which may also require hospitalization in an Intensive Care Unit (ICU) [9]. Moreover, other possible post-acute sequelae of SARS-CoV-2 infection, including myocarditis, loss of taste or smell, should not be underestimated [10,11]. According to the analysis of symptomatic COVID-19, in cases reported in The European Surveillance System (TESSy) by 10 European countries (Austria, Cyprus, Finland, Germany, Ireland, Italy, Luxembourg, Malta, Slovakia, and Sweden), 9611 hospitalizations and 84 deaths were registered in patients 0–17 years from weeks 32/2020 to 39/2021 [12]. The age group of 5 to 11 years old, which comprises 6.6% of the total population of these countries, increased from 3.8% to 11.2% of weekly cases during the same period, a threefold increase [13]. Therefore, it is important to vaccinate children, especially adolescents at high risk for other underlying diseases, which increase the possibility of developing the more severe or long forms of COVID-19. Vaccination of paediatric groups allows an increase in the vaccination coverage of the entire population and higher protection even for the most fragile subjects.

Although the safety profiles of mRNA COVID-19 vaccines have been evaluated in pre-authorization clinical trials, there is an urgent need for real-world data in order to better define their benefit/risk ratio, especially in this specific population, and also to reduce the parents’ hesitancy to vaccination. To this end, the present study aimed at evaluating the safety profile of mRNA COVID-19 vaccines by analysing the pharmacovigilance data of the European spontaneous reporting system database, EudraVigilance (EV).

## 2. Materials and Methods

### 2.1. Data Source

Data on Individual Case Safety Reports (ICSRs) of suspected Adverse Events Following Immunization (AEFIs) by mRNA COVID-19 vaccines (BNT162b2 and mRNA-1273) were retrieved from the EudraVigilance (EV) database. We selected all ICSRs with mRNA COVID-19 vaccines as suspected drugs and related to patients 3–11 years old (although the analysis was performed for the 5–11-year-old age group, EV only enables you to filter the 3–11-year-old age group) reported in EV from 1 January 2021 (gateway date) to 1 October 2022.

From each ICSR the following information was considered: a unique identifier number, the report type (spontaneous or non-spontaneous), the gateway receipt date, the primary source qualification (healthcare or non-healthcare professional), the primary source country (European or non-European Economic Area), the literature reference if available (some ICSRs could indeed be already published as a case report), the patient age group, the patient sex, and a list of AEFIs including their duration, outcome, and seriousness. Finally, a list of suspected and concomitant drugs was considered, including their therapeutic indication, use duration, dose, and route (when available) [14]. The reported AEFIs are coded as Lowest-Level Term (LLT) according to the Medical Dictionary for Regulatory Activities (MedDRA), the international medical terminology developed under the auspices of the International Council for Harmonisation of Technical Requirements for Pharmaceuticals for Human Use (ICH). Regarding the seriousness, a case is defined as serious by the International Council on Harmonization E2D guidelines, or rather when it results in death, is life-threatening, requires/prolongs hospitalization, results in persistent or significant disability/incapacity, is a congenital anomaly/birth defect, or results in some other clinically important condition. Furthermore, the outcomes are labelled as “Recovered/Resolved”, “Recovering/Resolving”, “Recovered/Resolved With Sequelae”, “Not Recovered/Not Resolved”, “Fatal”, and “Unknown”.

### 2.2. Descriptive Analysis

Data on case characteristics (including sex, report type, gateway receipt date, primary source qualification, country, and outcome), number and seriousness criteria of the reported adverse events, as well as a number of concomitant medicines other than mRNA COVID-19 vaccines were considered for each ICSR. We then provided a descriptive analysis of the demographic characteristics of ICSRs and the most relevant characteristics of the AEFIs, for cases related to both the BNT162b2 and mRNA-1273 vaccines. Moreover, we described the frequencies of AEFIs by System Organ Class (SOC). We also visualized the reporting trend for the study period. In order to perform a descriptive analysis as exhaustive as possible, we grouped the concomitant medicines according to the second level of the Anatomical Therapeutic Chemical (ATC) classification system. Moreover, in order to provide an analysis of ICSRs potentially suggestive of vaccination failure, we proceeded using the clinical criteria established for evaluating the efficacy of mRNA COVID-19 vaccines in preventing COVID-19 occurrence as reported in the European Assessment Reports. Specifically, we selected all ICSRs which reported the following LLT: “Drug ineffective” and/or “Vaccination failure” in combination with “COVID-19” and “fever” or “cough” or “shortness of breath” or “chills” or “muscle pain” or “loss of taste or smell” or “sore throat” or “diarrhoea” or “vomiting”. Except for “Drug ineffective”, “Vaccination failure”, and “COVID-19”, the other signs and symptoms were adapted to the MedDRA. Regarding to the reported outcome, our selection focused only on ICSRs with reported “death” or “hospitalization”.

### 2.3. Ethical Consideration

Because of data protection regulations, retrieved data included non-identifiable patient information, and free text narrative from the ICSRs was not available, hence no ethical review board permission was necessary.

## 3. Results

### 3.1. Characteristics of Individual Case Safety Reports (ICSRs)

During the study period, a total of 4838 ICSRs were related to paediatric subjects and the reported mRNA COVID-19 vaccine (BNT162b2 or mRNA-1273), as suspected. Out of 4838, 96.9% (N = 4689) of cases were related to the BNT162b2 vaccine, while 3.1% was the proportion related to the mRNA-1273 vaccine. Looking at ICSRs related to the BNT162b2 vaccine, we observed that the sex distribution was similar between male and female subjects (male subjects 49.7% and females 48.3%, respectively), and non-healthcare professional was the most frequent source of the selected ICSRs and came from the European Economic Area (EEA) (N = 3313, 70.7%); regarding the ICSRs involving the mRNA-1273 vaccine (N = 149), our data showed that the majority of ICSRs were related to female patients (63.8%), issued by healthcare professionals (N = 107, 71.8%), and came from the non-EEA (N = 98, 65.8%). All ICSRs were spontaneous (N = 4838, 100.0%). Almost all of the ICSRs included an mRNA vaccine without any other suspected drugs (N = 4793, 99.1%) or concomitant ones (N = 4786, 99.0%) (Table 1).

Starting from 1 January 2021 until 1 October 2022, as expected a marked increase in reported ICSRs was observed starting from the approval date of the extension of indication of use of the BNT162b2 vaccine for the 5–11 years age group (November 2021), with a pick recorded in March 2022 (Figure 2). We observed few cases as early as January 2021, with these ICSRs including AEFIs coded as “Wrong drug administered”, “Wrong drug”, and other LLTs that suggest a medical error (data not shown).

The overall 4838 paediatric ICSRs accounted for a total of 12,751 AEFIs reported (median = 2 AEFIs per ICSR; IQR = 1–3), of which 38.7% were serious, 24.2% were not serious, and for 37.1% (N = 4735) this information was not available. Similar to the total selected ICSRs, even for AEFIs related to the BNT162b2 vaccine, the distribution of non-serious and serious events was 61.8% and 38.2%, respectively. Looking at the mRNA-1273 vaccine, the distribution of serious and non-serious events was 60.8% and 39.2%, respectively. Considering the reported seriousness criteria, “other medically important condition” (N = 2828, 57.3%) and “caused or prolonged the hospitalization” (N = 1719, 34.8%) were the more commonly selected. Overall, the large majority of the AEFIs reported a positive outcome, such as “recovered/resolved”. This result was also confirmed focusing on the BNT162b2 vaccine (38.5%), while “recovering/resolving” was the most reported for the mRNA-1273 vaccine (50.9%) (Table 2).

The distribution of AEFIs was categorized by System Organ Class (SOC). The most common SOC was “General disorders and administration site conditions” (N = 3690, 28.9%), followed by “Nervous system disorders” (N = 2089, 16.4%), “Gastrointestinal disorders” (N = 1471, 11.5%), “Skin and subcutaneous tissue disorders” (N = 1111, 8.7%), “Infection and infestations” (N = 735, 5.8%), “Musculoskeletal and connective tissue disorders” (N = 643, 5.0%), “Respiratory, thoracic and mediastinal disorders” (N = 508, 4.0%), and “Injury, poisoning and procedural complications” (N = 447, 3.5%); the other SOCs accounted for less than 3% (Figure 3). More in detail, pyrexia, headache, vomiting, rash, COVID-19, pain in extremities, dyspnea, and product administered to patients of inappropriate age were the most common reported LLTs for each above-mentioned SOC (Table 3).

Irrespective of the suspected vaccine, other medicines were reported as concomitant in the ICSRs (Figure 4). The most common reported level ATC groups were “Respiratory system”: R03—drugs for obstructive airway diseases (N = 56, 22.0%), R06—antihistamines for systemic use (N = 31, 12.2%), and R01—nasal preparations (N = 7, 2.1%); “Nervous system”: N03—antiepileptics (N = 27, 10.6%) and N06—psychoanaleptics (N = 9, 3.5%); and “Antiinfectives for systemic use”: J07—vaccines (N = 20, 7.9%) and J01—antibacterials for systemic use (N = 5, 2.0%). Finally, from the analysis of ICSRs potentially suggestive of vaccination failure, among 4838 ICSRs only two cases matched our selection criteria (0.04%).

### 3.2. Multisystem Inflammatory Syndrome

Overall, we found that 20 ICSRs reported a Multisystem Inflammatory Syndrome as an AEFI. These were all spontaneous reports, mainly sent by healthcare professionals (N = 15) and referring to European Economic Area (N = 12). All reported MIS cases were related to BNT162b2, reported as a suspected drug. In only one ICSR, another suspected drug was reported, which was the influenza A virus H1N1 vaccine. This report described a “Recovered” MIS that occurred in a male subject also affected by COVID-19. In three cases, an ineffective drug was reported. The majority of ICSRs referred to female subjects (N = 12). Out of 20 MIS cases, one was fatal: this occurred in a female subject that also presented COVID-19 infection, acute respiratory distress syndrome, cardiomyopathy, and kidney transplant rejection. The outcome of MIS resulted as completely resolved in six ICSRs, of which only two ICSRs also reported the MIS duration (4 and 11 days). Moreover, six MIS were “Recovering” at the time of reporting, while the other four cases were “Not Resolved”. In the remaining three ICSRs, the adverse event outcome was “Unknown”. All MIS were categorized as serious events, resulting as life-threatening in four cases, requiring hospitalization in nine cases, or as other medically important conditions in seven cases.

## 4. Discussion

In this study, from 1 January 2021 to 1 October 2022, we analysed all mRNA COVID-19 vaccine-related Individual Case Safety Reports (ICSRs) of 5–11-year-old subjects reported in the EudraVigilance (EV) database.

A rapid positive trend in ICSR reporting was observed, peaking in January, according to the expansion of the European immunization campaign in this frail population. We analysed 4838 spontaneous ICSRs whose primary source was non-healthcare professionals. This result highly differs from what has been observed in the spontaneous reporting “history”, where healthcare professionals have been always the most common source. These findings could be due to both the unprecedented attention paid to COVID-19 vaccines especially from citizens and to greater awareness of patients about the spontaneous reporting of suspected adverse drug/vaccine events [15].

Looking at the distribution of ICSRs and vaccines available for children, as expected, almost all of them were related to the BNT162b2 vaccine (96.9%), and this finding might depend on its earlier marketing approval in children than the mRNA-1273 one [1,2].

With regard to sex distribution, several studies have shown sex differences in adverse events spontaneously reported in pharmacovigilance databases, showing a higher proportion of ICSRs related to female patients [16,17]. This sex difference in terms of ICSRs is still appreciable even in the case of COVID-19 vaccination cases as observed in recent studies, except for the cases of myocarditis and pericarditis which occur predominantly in young males [18]. However, regarding the paediatric population, as already observed in previous pharmacovigilance data analysis, our results showed that sex was almost equally distributed over the whole dataset, with a slight preponderance of males (49.3%) over females (48.7%) [19]. Moreover, stratifying by the vaccine, our analysis showed a similar sex distribution among ICSRs related to BNT162b2 (male 49.7%; female 48.3%), while a higher proportion of female sex was seen for the mRNA-1273 vaccine (male 35.6%; female 63.8%; *p* < 0.05), although they were outnumbered. 

As expected, considering the age group taken into consideration, almost all the ICSRs did not report other medicines besides the suspected vaccine, neither as suspected nor as a concomitant drug. We just identified 52 ICSRs (1.0%) with concomitant drugs reported, and, among these, our analysis showed a higher prevalence of the use of corticosteroids, bronchodilators, anti-leukotriene drugs, and antihistamines. These data could suggest that these cases were potentially related to patients affected by respiratory diseases; however, pharmacovigilance database descriptive analysis does not provide information about underlying and current diseases. Moreover, we cannot rule out that some of these concomitant drugs have been used to prevent potential allergic and anaphylactic reactions from vaccines, although it is not recommended [20].

Looking at the AEFI seriousness, our analysis showed that 38.7% of events were serious, and this finding could be related to the spontaneous reporting system of suspected adverse-event-related drugs and/or vaccines; indeed, there is a tendency to report serious adverse events. Moreover, if we considered the predominance of non-healthcare professionals as the primary source (53.8% of ICSRs were reported by non-HCPs) together with the hesitancy and fear of vaccination in children, this finding it is not surprising [21]. 

Moreover, in line with these findings, A.J. Avery et al. and L. Rolfes et al. also demonstrated that consumers more frequently reported many serious adverse events that affected their quality of life. Furthermore, consumers’ descriptions were more detailed and allowed the identification of higher potential signals, which would not have been detected when received from healthcare professionals, even though their reports contained more useful clinical information [22,23].

In terms of reported outcomes, in most the cases the event was classified as fully resolved or as being resolved. As already reported in the mRNA COVID-19 vaccines’ Risk Management Plan (RMP) and the Summary of Product Characteristics (SmPC), the most common adverse events are usually mild or moderate and get better within a few days after vaccination. Specifically, systemic events such as headache, muscle and joint pain, vomiting, diarrhoea, and fever resolved within a median duration of 1 to 2 days after onset [1,23]. In this regard, Molteni et al. showed in their prospective longitudinal cohort study that post-vaccination local and systemic side effects resolved within a few days (3 days in most cases) [2]. However, it was not possible to retrieve information on the duration of the event from our dataset. 

Looking at the type of events, pyrexia was the most commonly reported AEFI belonging to the SOC “General disorders and administration site conditions”, headache within “Nervous system disorders”, vomiting within “Gastrointestinal disorders”, and rash within “Skin and subcutaneous tissue disorders”, as expected. These findings have already been documented in the pre-authorization trials of Moderna and Pfizer-BioNTech vaccines in children aged 5 to 11 years, which demonstrated that short-term systemic adverse events (such as pyrexia, headache, and vomiting) and local reactions (e.g., rash) occurred more frequently [24,25]. Our data are also in line with other studies conducted after the introduction of mRNA COVID-19 vaccines in this category. A pharmacovigilance study, conducted on data from the Vaccine Adverse Events Reporting System (VAERS), found 7578 reports of AEFIs for children ages 5 to 11 years who received the BNT162b2 vaccine. Among the most reported AEFIs included pyrexia (541, 7.3%), headache (465, 6.3%), vomiting (541, 7.3%), and rash (311, 4.2%) [26]. Additionally, in a real-world setting, Bloise et al. recently evaluated the safety of the mRNA COVID-19 vaccines through a prospective, cross-sectional study. Specifically, the authors reported that the systemic adverse effects reported 24–48 h after the first dose were headache (8.1%) and fever (2.9%) [27]. 

Regarding the focus on the ICSRs suggestive of potential vaccination failure, among 4838 ICSRs only two cases met our selection criteria (0,04%), confirming the high efficacy already evaluated in the registration trials.

Finally, we focused our attention on ICSRs reporting Multisystem Inflammatory Syndrome in children (MIS-c) reported as AEFIs. The onset of this syndrome following immunization as well as after Severe Acute Respiratory Syndrome COronaVIrus 2 infection (SARS-CoV-2) is considered a topic of interest and debate [28,29,30,31]. This syndrome results in a systemic hyperinflammatory state in patients <21 years of age, which requires hospitalization. Signs and symptoms include fever experienced >24 h, involvement of >2 organ systems, and >1 among specific laboratory results, such as elevated C-reactive protein (CRP), erythrocyte sedimentation rate, fibrinogen, procalcitonin, D-dimer, ferritin, lactate dehydrogenase, interleukin-6, neutrophils, reduced lymphocytes, or reduced albumin [29]. The relationship between SARS-CoV-2 exposure, COVID-19 vaccination, and MIS-c is still debated since it is unclear the major contribution of the COVID-19 vaccination or infection to the onset of this serious syndrome [32]. In October 2021, the Pharmacovigilance Risk Assessment Committee (PRAC) considered the evidence of a possible link between MIS-c and mRNA COVID-19 vaccines as insufficient [33]. As emerged also from our analysis, this adverse event is rarely observed following COVID-19 vaccination. Contrary to the results of a recent French post-authorization pharmacovigilance study according to which a male predominance emerged, in our analysis, the majority of MIS-c cases occurred in female subjects [34]. It would be interesting to verify whether the different sex distribution can be related to the different paediatric age groups considered (5–11 vs. 12–17 years old). To date, considering the uncertain link with mRNA vaccination, MIS-c remains the most severe life-threatening clinical entity associated with paediatric SARS-CoV-2 infection [34].

## 5. Strengths and Limitations

This study has both the typical strengths and limitations of safety analyses using the pharmacovigilance database. These kinds of data reflect both real-life events and real-life drug use, including drug use patterns that, for ethical reasons, cannot be investigated in clinical trials. Moreover, it might be possible to detect rare adverse events and to observe medication errors, which cannot appear in pre-authorization clinical trials. Finally, EudraVigilance is one of the largest pharmacovigilance databases, collecting heterogeneous information from different countries and populations. In our specific analysis, we were able to observe mRNA COVID-19 vaccine-related adverse events in a population group that is typically under-represented in pre-authorization clinical trials. Although these strengths exist, several limitations need to be considered when interpreting this type of result. Under-reporting is highly likely in a spontaneous report system and reported adverse events may only be the tip of the iceberg, especially in paediatric patients [35,36,37]. Furthermore, ICSRs vary in quality and completeness. They often lack useful quantitative data (e.g., the antibody titre) to better define each case. The lack of additional useful information (e.g., the severity of the underlying illness, a causality assessment, a detailed description of the adverse events, or a follow-up) makes it impossible to infer a causal link between medicines and adverse events [38]. Regarding the severity of the events, more severe and unexpected medical occurrences are probably more likely to be reported by both physicians and patients than minor ones, especially when they happen soon after vaccination, even though they can be coincidental and linked to other causes (e.g., an underlying illness or concomitant medications). Given the limitation of the data source and the descriptive nature of our analysis, interpreting these results requires caution.

## 6. Conclusions

The current study provides an overview of spontaneous reports of adverse events in paediatric subjects vaccinated with an mRNA COVID-19 vaccine (Pfizer-BioNTech or Moderna). Results of this post-marketing analysis suggest a favourable risk–benefit profile for all mRNA COVID-19 vaccines in this paediatric sub-population, supporting their use in children. Given the above-mentioned limitations of this analysis, further studies need to be performed to investigate the safety profile of the mRNA COVID-19 vaccines. Considering multiple SARS-CoV-2 variants circulating internationally with increased transmissibility and that the benefits of COVID-19 vaccination outweigh the known and potential risks, the use of COVID-19 vaccines can be promoted in the age group of 5–11 years.

## Figures and Tables

**Figure 1 vaccines-11-00401-f001:**
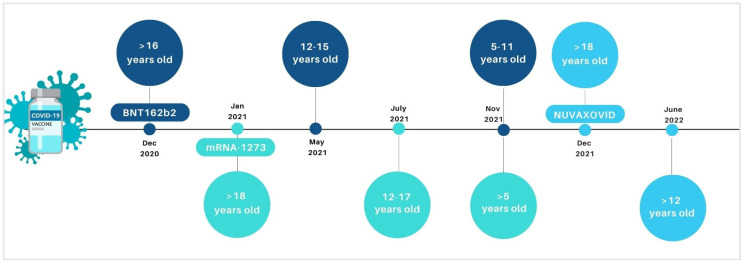
Timeline on the extension of indication for the COVID-19 vaccines in paediatric age groups by European Medicine Agency.

**Figure 2 vaccines-11-00401-f002:**
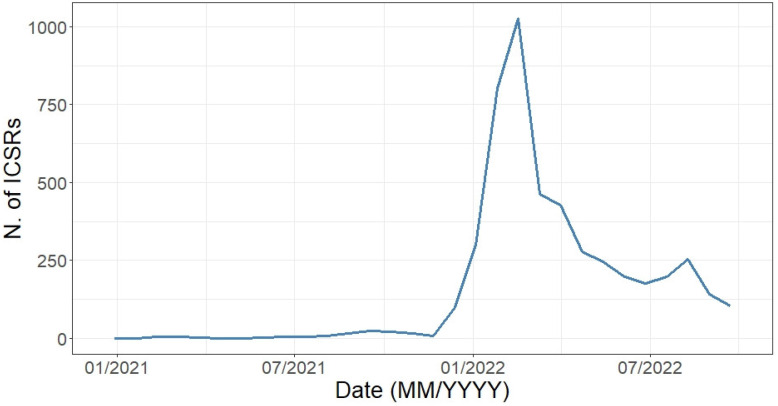
Distribution of Individual Case Safety Reports having an mRNA COVID-19 vaccine as suspect drugs by month (1 January 2021–1 October 2022).

**Figure 3 vaccines-11-00401-f003:**
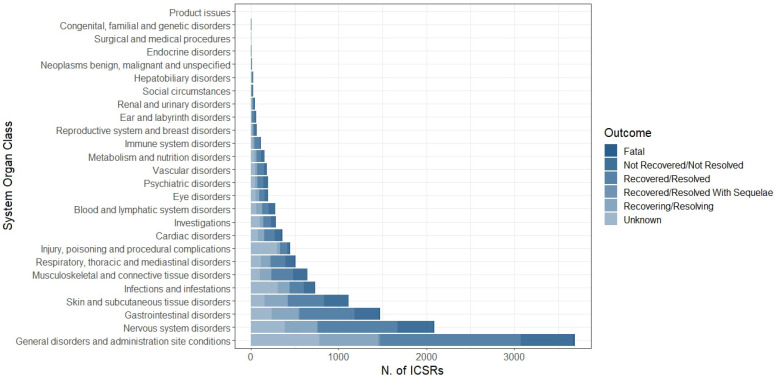
Distribution of System Organ Class (SOC) related to each AEFI (N = 12,751) reported in overall Individual Case Safety Reports.

**Figure 4 vaccines-11-00401-f004:**
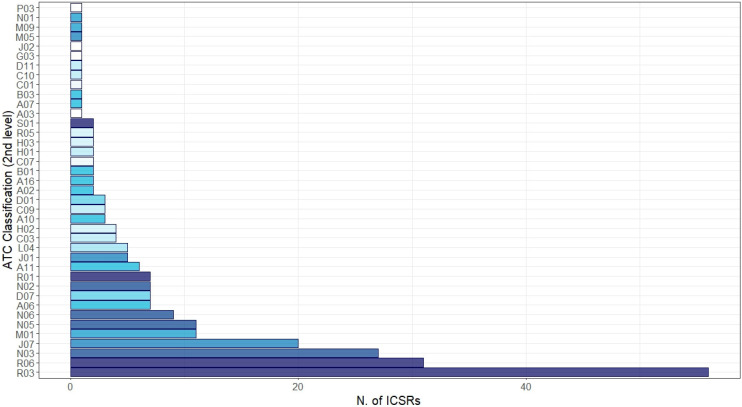
Distribution of medicines reported as concomitant with mRNA COVID-19 vaccines grouped by therapeutic group (2nd level of Anatomical Therapeutic Chemical (ATC)). Legend: R03: drugs for obstructive airway disease; R06: antihistamines for systemic use; N03: antiepileptics; J07: vaccines; M01: anti-inflammatory and antirheumatic product; N05: physcholeptics; N06: psychoanaleptics; A06: drugs for constipation; D07: corticosteroids, dermatological preparations; N02: analgesics; R01: nasal preparations; A11: vitamins; J01: antibacterials for systemic use; L04: immunosuppressants; C03: diuretics; H02: corticosteroids for systemic use; A10: drugs used in diabetes; C09: agents acting on the renin–angiotensin system; D01: antifungals for dermatological use; A02: drugs for acid-related disorders; A16: other alimentary tract and metabolism products; B01: antithrombotic agents; C07: beta blocking agents; H01: pituitary and hypothalamic hormones and analogues; H03: thyroid therapy; R05: cough and cold preparations; S01: ophthalmologicals; A03: drugs for functional gastrointestinal disorders; A07: antidiarrheals, intestinal anti-inflammatory/anti-infective agents; B03: anti-anaemic preparations; C01: cardiac therapy; C10: lipid modifying agents; D11: other dermatological preparations; G03: sex hormones and modulators of the genital system; J02: antimycotics for systemic use; M05: drugs for treatment of bone diseases; M09: other drugs for disorders of the musculo-skeletal system; N01: anaesthetics; P03: ectoparasiticides, incl. scabicides, insecticides, and repellents.

**Table 1 vaccines-11-00401-t001:** Demographic characteristics of Individual Case Safety Reports (ICSRs) involving BNT162b2 and mRNA-1273 vaccines reported in the EudraVigilance spontaneous reporting system from 1 January 2021 to 1 October 2022.

	BNT162b2 ICSRs ^a^ N = 4689 (96.9%)	mRNA-1273 ICSRs ^a^ N = 149 (3.1%)	All ICSRs ^a^ N = 4838 (100.0%)
**Sex**			
Male	2330 (49.7)	53 (35.6)	2383 (49.3)
Female	2263 (48.3)	95 (63.8)	2358 (48.7)
NA	96 (2.0)	1 (0.6)	97 (2.0)
**Primary Source Qualification**			
Non-Healthcare Professional	2558 (54.5)	42 (28.2)	2600 (53.8)
Healthcare Professional	2131 (45.5)	107 (71.8)	2238 (46.2)
**Primary Source Country for Regulatory Purposes**			
European Economic Area	3313 (70.7)	51 (34.2)	3364 (69.5)
Non-European Economic Area	1376 (29.3)	98 (65.8)	1474 (30.5)
**Report Type**			
Spontaneous	4689 (100.0)	149 (100.0)	4838 (100.0)
Non-Spontaneous	-	-	-
**Suspected drug(s) other than the COVID-19 vaccine**			
0	4650 (99.2)	143 (96.0)	4793 (99.1)
1	38 (0.7)	6 (4.0)	44 (0.8)
≥2	1 (0.1)	-	1 (0.1)
**Concomitant drug(s)**			
0	4637(99.0)	149 (100.0)	4786 (99.0)
1	24 (0.5)	-	24 (0.5)
2	-	-	-
≥3	28 (0.5)	-	28 (0.5)

^a^ Individual Case Safety Reports (ICSRs).

**Table 2 vaccines-11-00401-t002:** Characteristics of Adverse Events Following Immunization (AEFIs).

	BNT162b2 AEFIs ^a^ N = 12450 (97.6%)	mRNA-1273 AEFIs ^a^ N = 301 (2.4%)	All AEFIs ^a^ N = 12751 (100.0%)
**AEFI Seriousness**			
Serious	4751 (38.2)	183 (60.8)	4934 (38.7)
Not Serious	7699 (61.8)	118 (39.2)	7817 (61.3)
**AEFI Seriousness Criteria**			
Other Medically Important Condition	2743 (57.8)	85 (46.4)	2828 (57.3)
Caused/Prolonged Hospitalization	1625 (34.2)	94 (51.5)	1719 (34.8)
Disabling	163 (3.4)	1 (0.5)	164 (3.3)
Life Threatening	134 (2.8)	2 (1.1)	136 (2.8)
Results in Death	86 (1.8)	1 (0.5)	87 (1.8)
**AEFI Outcome**			
Recovered/resolved	4798 (38.5)	75 (24.9)	4873 (38.2)
Not recovered/not resolved	2387 (19.2)	41 (13.6)	2428 (19.0)
Recovering/resolving	2209 (17.7)	153 (50.9)	2362 (18.5)
Fatal	86 (0.7)	1 (0.3)	87 (0.7)
Recovered with sequelae	83 (0.7)	3 (1.0)	86 (0.7)
Unknown	2887 (23.2)	28 (9.3)	2915 (22.9)
**Median AEFIs per ICSR (IQR)**	2 (1–3)	2 (1–3)	2 (1–3)

**^a^** Adverse Events Following Immunization (AEFIs). The total number of reported Adverse Events Following Immunization is different from the total number of the analysed ICSRs and this apparent discrepancy is explained by the fact that one ICSR could report more than one adverse event.

**Table 3 vaccines-11-00401-t003:** Distribution of Lowest-Level Term (LLT) (at least accounted for 3% of all AEFIs) belonging to the top 8 System Organ Classes (SOCs).

System Organ Class and Lowest-Level Term	N. of Adverse Events (%)
**General disorders and administration site** **conditions**	**3690 (100.0)**
Pyrexia	899 (24.4)
Injection site pain	436 (11.8)
Fatigue	241 (6.5)
Drug ineffective	212 (5.7)
Malaise	169 (4.6)
Chest pain	164 (4.4)
Vaccination site pain	129 (3.5)
**Nervous system disorders**	**2089 (100.0)**
Headache	732 (35.0)
Syncope	254 (12.2)
Dizziness	216 (10.3)
Seizure	142 (6.8)
Loss of consciousness	81 (3.9)
**Gastrointestinal disorders**	**1471 (100.0)**
Vomiting	508 (34.5)
Nausea	286 (19.4)
Abdominal pain	227 (15.4)
Diarrhoea	181 (12.3)
Abdominal pain upper	74 (5.0)
**Skin and subcutaneous tissue disorders**	**1111 (100.0)**
Rash	307 (27.6)
Urticaria	187 (16.8)
Pruritus	112 (10.1)
Erythema	72 (6.5)
Hyperhidrosis	43 (3.9)
Rash pruritic	33 (3.0)
**Infections and infestations**	**735 (100.0)**
COVID-19	329 (44.8)
Influenza	38 (5.2)
Suspected COVID-19	38 (5.2)
Herpes zoster	37 (5.0)
Nasopharyngitis	26 (3.5)
Appendicitis	25 (3.4)
**Musculoskeletal and connective tissue disorders**	**643 (100.0)**
Pain in extremity	194 (30.2)
Myalgia	135 (21.0)
Arthralgia	98 (15.2)
Limb discomfort	43 (6.7)
Muscular weakness	23 (3.6)
**Respiratory, thoracic, and mediastinal disorders**	**508 (100.0)**
Dyspnoea	125 (24.6)
Cough	104 (20.5)
Oropharyngeal pain	61 (12.0)
Epistaxis	39 (7.7)
Rhinorrhoea	24 (4.7)
**Injury, poisoning, and procedural complications**	**447 (100.0)**
Product administered to patient of inappropriate age	80 (17.9)
Vaccination failure	75 (16.8)
Off-label use	51 (11.4)
Fall	30 (6.7)
Product use issue	19 (4.3)
Inappropriate schedule of product administration	17 (3.8)
Overdose	17 (3.8)
Contusion	16 (3.6)
Vaccination error	15 (3.4)

## Data Availability

Publicly available datasets were analysed in this study. These data can be found here: www.adrreports.eu (accessed on 2 October 2022).

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
