# Peer review of "Safety Monitoring of mRNA COVID-19 Vaccines in Children Aged 5 to 11 Years by Using EudraVigilance Pharmacovigilance Database: The CoVaxChild Study"

_vaccines, 2023, doi:10.3390/vaccines11020401_

Round 1

Reviewer 1 Report

Although the subject of this article is very interesting, my main concern is about the lack of use of appropriate methodological tools to support the authors conclusion about good safety profile of the vaccines. How could the authors conclude that there is no safety signals using only simple summary percentages? Although necessary to decribe the available data, they remain insuffisant to reach a sufficient level of evidence to formulate such a conclusion.

The rate AEFIs without available information is very high 37.1% ! What if we consider different scenarios, such as the extreme one of all of them were serious?

Line 56: the use of photos in a legend is probably not appropriate

Line 58 : replace Medicines by Medicine

Line 84 : replace special by specific

Figure 2 : replace Date, N., gen, lug by clear and appropriate words

Author Response

Dear Reviewer, we appreciate you taking an interest in the topic of our manuscript. Thank you for the time and effort you invested in giving us feedback on our analysis.   Regarding your doubts about the AEFI's "Not Available" seriousness, a manipulation data error was detected after a comprehensive examination of the dataset. Actually, no "Not Available" seriousness was observed in our data, so we updated Table 2 and the results (see lines: 195-200).   We expanded the limitation section (see lines: 439-449).   As you observed, given the descriptive nature of our analysis, no safety signal can be identified. Therefore, we made changes to the conclusion section to emphasize this problem.   We reported your recommended corrections (see lines: 63 (Figure 1), 65, 92, 182 (Figure 2)), and an english revision was done.

Reviewer 2 Report

we read with interest the article by Zinzi et al titled "Safety monitoring of mRNA COVID-19 vaccines in children aged 5 to 11 years by using EudraVigilance pharmacovigilance  database: the CoVaxChild Study" where the authors are assessing the biosafety of the mRNA vaccines in the pediatric population aged 5-11 utilizing the data based "the European spontaneous reporting system database EudraVigilance (EV) in the period from January 1st, 2021, to Oc-18 tober 1st, 2022."

The study is of extreme importance and  highly timely few comments are highlighted:

Major comments,

while the resssults are presented, one need to be cautious in interpretting these results and this should be highlighted in the limitation section as the data do not indicate if any of the subjcets were given a booster shot or not.

second,  the data is descriptive innature, thus one can not infer if any ofthe adverse effcets were due to the combination of the otherdrugs that some subjects were taking (there is a slim percentage that this can occur).

third, the work would necessitate soem quantitative data related to antibody titer to recaah a finaal verdict about teh efficacy of the mRNA vaccines. (please indicate this in the limitation section)

forth, a future direction section that can be targetting public health sector on a follow up of these subjcets if they encounter other infection due to other Covid exposure etc.

Minor comments:

please define: ICSR in the abstrcat

Author Response

Dear Reviewer, we would like to thank you for careful and thoughful reading of our manuscript and for the thoughful comments and suggestions, which help us to improve the quality of this manuscript.   As you observed, given the limitation of the data source and the descriptive nature of our analysis, interpreting our results requires caution, so we expanded the limitation section (see lines: 439-449).   We defined "ICSRs" in the abstract.

Reviewer 3 Report

The manuscript refers to the analysis of vaccine pharmacovigilance of mRNA vaccine in pediatric patients. The manuscript resumes an exciting analysis of the pediatric population. The manuscript is original, and the analysis has been appropriately conducted. The discussion requires some more details and new references. The addition of strengths and limitations of the study is appreciated.

Author Response

Dear Reviewer, we appreciate you taking an interest in the topic of our manuscript. Thank you for your time and effort in giving us feedback on our analysis.   We expanded the limitation section, and an english revision was done.

Round 2

Reviewer 1 Report

Although it is understandable to have imperfect data, the authors should explain why there is so much missing and unknown data (it could be that these are cases under consideration?)

Please discuss whether this affects the results (e.g. do we know if they are unimportant cases?), and how more detail could be obtained?

Author Response

Dear Reviewer,

Thank you for your thoughtful feedback, which also provides us the opportunity to offer here further information that can be helpful to address your questions. The spontaneous reporting systems are affected by limitations that are mainly related to the underreporting and inaccuracy or incompleteness of information. Considering these intrinsic limitations, we cannot rule out the presence of information that were not listed in ICSRs, such as those related to specific risk factors (patients' age, concomitant diseases) that may have contributed to the occurrence of the adverse events.

Moreover, as we reported in the Table 1, the whole ICSRs were “spontaneous” and more than half of the ICSRs (N=2,558; 54.5%) were reported by non-healthcare professionals. As a result, the “passive” nature of these pharmacovigilance data and the main source qualification cannot guarantee the completeness of the data. We might collect more data through an active surveillance system, stimulating healthcare professionals with individual input.

Regarding the missing data, we found unknown information on sex (N=96; 2.0%) and outcome (N=2,887; 23.3%); about the outcomes, they refer to each adverse event, whereas in a case-by-case analysis we can see the outcome that refers to each individual case report. So, in our opinion, these missing data don’t affect the results.